# Solution-Processed SnO_2_ Quantum Dots for the Electron Transport Layer of Flexible and Printed Perovskite Solar Cells

**DOI:** 10.3390/nano12152615

**Published:** 2022-07-29

**Authors:** Muhammad Salman Kiani, Zhandos T. Sadirkhanov, Alibek G. Kakimov, Hryhorii P. Parkhomenko, Annie Ng, Askhat N. Jumabekov

**Affiliations:** 1Department of Physics, Nazarbayev University, Nur-Sultan 010000, Kazakhstan; muhammad.kiani@nu.edu.kz (M.S.K.); zhandos.sadirkhanov@nu.edu.kz (Z.T.S.); alibek.kakimov@gmail.com (A.G.K.); hryhorii.parkhomenko@nu.edu.kz (H.P.P.); 2Department of Electrical and Computer Engineering, Nazarbayev University, Nur-Sultan 010000, Kazakhstan; annie.ng@nu.edu.kz

**Keywords:** perovskites, solar cells, low-temperature, solution-processed, quantum dots, electron transport layer

## Abstract

Flexible and printed perovskite solar cells (PSCs) fabricated on lightweight plastic substrates have many excellent potential applications in emerging new technologies including wearable and portable electronics, the internet of things, smart buildings, etc. To fabricate flexible and printed PSCs, all of the functional layers of devices should be processed at low temperatures. Tin oxide is one of the best metal oxide materials to employ as the electron transport layer (ETL) in PSCs. Herein, the synthesis and application of SnO_2_ quantum dots (QDs) to prepare the ETL of flexible and printed PSCs are demonstrated. SnO_2_ QDs are synthesized via a solvothermal method and processed to obtain aqueous and printable ETL ink solutions with different QD concentrations. PSCs are fabricated using a slot-die coating method on flexible plastic substrates. The solar cell performance and spectral response of the obtained devices are characterized using a solar simulator and an external quantum efficiency measurement system. The ETLs prepared using 2 wt% SnO_2_ QD inks are found to produce devices with a high average power conversion efficiency (PCE) along with a 10% PCE for a champion device. The results obtained in this work provide the research community with a method to prepare fully solution-processed SnO_2_ QD-based inks that are suitable for the deposition of SnO_2_ ETLs for flexible and printed PSCs.

## 1. Introduction

With the tremendous improvement in the power conversion efficiency (PCE) up to 25.7% in the matter of a decade, hybrid perovskite solar cells (PSCs) made their mark in the electronic industry [1,2,3,4]. Even though some challenges such as device stability still need to be addressed, there has been a growing interest in scaling up the PSC fabrication process. Most high-efficiency PSCs are currently manufactured using the spin-coating method because of the better control over the film thickness, quality, and reproducibility of the device functional layers as well as the possibility of applying different spin-coating techniques such as dynamic spinning, hot-casting, anti-solvent annealing, gas and vacuum-quenching, etc. [5,6]. In comparison to other solution-processed techniques, spin-coating is an easy method to fabricate high-quality perovskite films [7]. One of the main limitations of the spin-coating method is its non-scalability [8]. Hence, various scalable methods including screen printing, blade coating, inkjet printing, and slot-die coating have been developed to fabricate PSCs [9,10,11,12,13,14]. The printing methods provide an appealing strategy for large-scale and high-throughput fabrication and can help to accelerate the commercialization of PSC technology [15,16,17,18].

Slot-die coating is one of the more appealing scalable coating methods due to its ability to produce solution-processed thin films with a high uniformity across large areas, its ability to coat thin films at a higher speed, its lesser material waste, and its ability to be integrated into both sheet-to-sheet (S2S) and roll-to-roll coating (R2R) systems. This affords the fabrication of PSCs on both rigid and flexible substrates [19,20]. Flexible and printed PSCs are of high interest due to their lightweight nature, portability, and low-manufacturing cost, along with the possibility of integrating them into other technologies, which promises many potential lucrative applications of this technology in wearable electronics, the internet of things (IoT), smart buildings, the automobile industry, etc. Flexible devices can be achieved by replacing traditional glass substrates with plastic substrates, such as polyethylene terephthalate (PET) or polyethylene naphthalate (PEN), with a thin layer of transparent conducting oxide (TCO). The TCO layer is typically made of indium-doped tin oxide (ITO). However, other alternatives also exist [21,22,23].

To successfully fabricate flexible PSCs, the three main semiconducting device functional layers, which are the electron transport layer (ETL, n-type), the perovskite photo-absorber layer (i-type), and the hole transport layer (HTL, p-type), must be processed at temperatures below the glass transition temperature of the employed plastic substrates (usually < 150 °C) [24,25,26]. The ETL is responsible for the extraction of electrons from the perovskite photo-absorber layer and then for their efficient transportation to the TCO layer. Mostly, metal oxides such as ZnO, TiO_2_, or SnO_2_ are used as ETL [27,28]. However, there are other alternatives that can be based on ternary metal oxides and organic materials (e.g., BaSnO_3_, Zn_2_SnO_4_, PEDOT:PSS, PEDOT:PSS:MoS_2_, etc.) [29,30]. By covering the surface of TCO, ETL prevents the direct contact between the TCO and perovskite layers [31]. The selection of an efficient ETL requires a material with a good optical transmittance (in the visible range), a high electron mobility, a low production cost, and the energy levels (conduction and valence band energies) that form a good type-2 junction with the energy levels of the chosen perovskite material [32]. Among all the candidates, SnO_2_ stands out as one of the best candidates for ETL due to its wide optical band gap (3.6–4.0 eV), deep conduction band, better transparency, high electron mobility (~240 cm^2^ V^−1^ s^−1^), long carrier diffusion length, excellent chemical stability, and ease of low-temperature preparation (via solution processing) [33,34,35,36]. Due to its solution processibility, SnO_2_ can be printed and bring advantages such as a low production cost, flexibility, easy scalability, and additive manufacturing processes within reach [37,38,39].

In 2018, T. Bu et al. reported slot-die coated flexible PSCs with an ETL made of commercially available SnO_2_ NPs (15% of colloidal dispersion in H_2_O, Alfa Aesar). The resultant devices, in which a mixed cation and mixed anion perovskite (Cs_0.05_(FA_0.85_MA_0.15_)_0.95_Pb(I_0.85_Br_0.15_)_3_) are employed as a photoactive layer and Spiro-OMeTAD is employed as an HTL, showed PCE values around 18% [40]. In 2020, Vijayaraghavan et al. reported the use of solution-processed SnO_2_ quantum dots as an ETL for PSCs. The ETL was prepared in an aqueous medium using DI water, SnCl_2_·2H_2_O, and thiourea as precursors. PCE values of around 13.5% have been achieved at an annealing temperature of 180 °C, and the devices are fabricated on glass substrates via the spin-coating method [41]. In 2021, Ashina et al. deposited the SnO_2_ ETLs using a dip-coating method on glass substrates with a fluorine-doped tin oxide (FTO) layer as the TCO. The authors deposited multiple layers (1–4) of a SnO_2_ precursor solution and achieved PCE values around 3.2%. The annealing temperature for the ETL was set to 150 °C [1]. While the aforementioned PCE values obtained with the SnO_2_ ETLs are appreciable, the annealing temperatures used in these studies, however, are still too high from the flexible PSC fabrication perspective. In 2021, Ren et al. reported a fabrication of PSCs on glass/FTO substrates using the blade coating method, in which the ETLs of devices are made of SnO_2_ QDs with an annealing temperature of 200 °C. The authors achieved devices with PCE values around 21.6%. The obtained devices are fabricated by using a mixed perovskite consisting of FAI:MABr:MACl:PbBr_2_:PbI_2_ (molar ratio: 1.1:0.2:0.5:0.2:1.2) in DMF:DMSO solvent and by employing an air-knife for the deposition of the perovskite layer [42]. Using SnO_2_ QDs instead of NPs provides the advantage of having a thin, dense, and uniform film of ETL. QDs have a wider bandgap (about 4.20 eV) compared to NPs, and this enhances the optical transmittance of the resulting ETL [43]. The main issue that limits the use of SnO_2_ ETL for flexible devices is its annealing temperature, which has been found > 150 °C, while the typical flexible substrates such as PET and PEN usually have temperature limitations in terms of their structural integrity [40,44,45,46]. One of the possible solutions for this could be the use of already crystalline SnO_2_ QDs for the solution-processed deposition of SnO_2_ ETLs. Here, the use of already crystalline SnO_2_ QDs would not only result in crystalline SnO_2_ ETLs but would also allow for the avoidance of the high-temperature processing step, which is crucial for devices based on plastic substrates. The low-temperature and solution-processed deposition of SnO_2_ ETL is advantageous from the perspective of flexible and printed optoelectronics.

In this work, a method to employ SnO_2_ quantum dots, synthesized via a solvothermal method, to prepare ETLs of flexible PSCs fabricated on PET/ITO substrates is demonstrated. For the synthesis of SnO_2_ quantum dots, tin chloride and ethanol are used as the solute and solvents, respectively. The obtained quantum dots are crystalline with sizes ~2.4 nm [47]. The obtained SnO_2_ QDs are used to prepare aqueous inks with different SnO_2_ QD concentrations to slot-die coat ETLs on PET/ITO substrates with a low annealing temperature. The obtained ETLs on PET/ITO substrates are then used to fabricate flexible and printed PSCs by also using the slot-die coating method. The metallic contacts (gold electrodes) of the devices are deposited using a thermal evaporator and a shadow mask. The solar cell performances of the devices are compared to rule out the best ETL ink composition. A statistical analysis of the device performance parameters is carried out, and the performance of the champion device is discussed. The findings of this study can be useful in the synthesis and use of SnO_2_ QDs for application in printed and flexible optoelectronics.

## 2. Materials and Methods

### 2.1. Materials

Methylammonium iodide (MAI, 99.995%) was purchased from GreatCell Solar. Tin chlo-ride dihydrate (SnCl_2_·2H_2_O, 98%), lead iodide (PbI_2_, 99%), chlorobenzene (CB, 99.80%), 4-tert-Butylpyridine (4tBp, 98%), bis(trifluoromethane)sulfonimide lithium salt (Li-TFSI, 99%), acetonitrile (ACN, 99.80%), methylamine solution (MA, 33 wt% in ab-solute ethanol), and acetone (99.80%) were purchased from Merck. 2,2′,7,7′-Tetrakis-(*N*,*N*-di-4-methoxyphenylamino)-9,9′-spirobifluorene (Spiro-MeOTAD, 99.50%) and potassium hydroxide pellets (KOH, 85%) were purchased from Lumtec and ACS reagent, respectively. 2-Propanol (IPA, 99.80%) and ethanol (96.3%) were purchased from a local supplier. Gold (Au, 99.99%) was purchased from Kurt. J. Lesker. All the chemicals were used as received without any treatment.

### 2.2. SnO_2_ QDs Synthesis

SnO_2_ QDs were prepared by dissolving 1 mol of SnCl_2_·2H_2_O in 10 mL of ethanol in the presence of oxygen. The mixture of precursors was vigorously stirred for 4 h using a magnetic stirrer. The obtained QD solution was then transferred into a 50 mL Teflon liner in a stainless-steel autoclave (Toption). The solvothermal treatment was carried out at 150 °C in an oven for 24 h [48]. After this, the obtained solution was centrifuged to precipitate the SnO_2_ QDs. The obtained precipitate (SnO_2_ QDs) was washed consecutively with DI water, acetone, and ethanol. Then, the SnO_2_ QDs were dried at 80 °C in the oven for 12 h. A flow chart in Figure 1 shows the process for the synthesis of SnO_2_ QDs to prepare SnO_2_ QD inks.

### 2.3. Ink Preparations

The aqueous SnO_2_ QD inks were prepared using freshly synthesized SnO_2_ QDs. For this, the dried powder of SnO_2_ QDs was ground and redispersed in DI water (at different wt%) with the help of 1 M KOH (in water) solution as a pH mediator (see Figure 1). The obtained colloidal solution was then filtered using a 0.45 μm-sized PTFE filter and further diluted with DI water to obtain the final ink solutions, with the concentrations of SnO_2_ QDs being 1, 1.5, 2, 2.5, and 3 wt%. The perovskite (methylammonium lead iodide, MAPbI_3_) ink was prepared using PbI_2_ and MAI as precursors, while MA and ACN were used as solvents. For this, 1 mol of PbI_2_ and MAI were dissolved in 700 µL of MA by stirring at 70 °C and 460 rpm for 1 h using a hot plate with a magnetic stirrer. Next, 700 µL of ACN was added to the solution and stirred at room temperature for an additional 3 h. The obtained final perovskite ink solution was then filtered using a 0.45 μm-sized PTFE filter and was ready for use. The HTL ink was prepared by simply dissolving 45 mg of Spiro-MeOTAD in 1 mL of CB using a vortex shaker. After 1 h of shaking, 28.8 µL 4tBp and 17.5 µL lithium salt solution (520 mg mL^−1^ of Li-TFSI in ACN) were added to the Spiro-MeOTAD/CB solution. The final HTL ink solution was then filtered using a 0.45 μm-sized PTFE filter for further use.

### 2.4. Device Fabrication

For the fabrication of flexible and printed PSCs, a freshly cleaned PET/ITO substrate was placed onto the chuck of a slot-die coater (VectorSC, FOM Technologies, Copenhagen, Denmark) and fixed using vacuum suction to keep the substrate perfectly flat on the surface of the chuck. The PET/ITO substrates were cleaned by (1) washing them consecutively in DI detergent water, clean DI water, acetone, and IPA for 10 min in each media using an ultrasonic bath, (2) drying them with the flow of compressed air, and (3) subjecting them to UV-Ozone treatment for 10 min. The SnO_2_ QDs inks were printed on the PET/ITO substrates using a clean slot-die head. Then, the substrates were annealed in the oven at 140 °C for 30 min. This step is necessary to remove the remaining water from the ETL film prior to the deposition of the perovskite layer on top of it. Once the substrates were cooled down to room temperature, they were placed onto the chuck of the slot-die coater again, and the perovskite ink was printed on top of the ETL using a clean slot-die head. Then, the substrates were annealed again in the oven at 100 °C for 10 min to crystallize the perovskite layer. Next, the HTL ink was printed on top of the perovskite layer using a clean slot-die head. All of the device functional layers were printed in an ambient environment at room temperature (17 °C) and a relative humidity of 20–24%. After depositing all the functional layers, the substrates were left in the atmosphere of oxygen at room temperature for 4 h (oxygen annealing). This step helps to enhance the hole transporting properties of the Spiro-MeOTAD HTL. Finally, the 80 nm-thick Au electrodes were deposited using a vacuum thermal evaporator (Mini SPECTROS, Kurt J. Lesker, USA) and a shadow mask. Overall, the devices have the following structure: PET/ITO/SnO_2_/MAPbI_3_/Spiro-MeOTAD/Au. Figure 2 represents a schematic diagram of the device fabrication process and an energy level diagram for the fabricated PSCs [49,50]. The reference devices without ETL were also fabricated using the exact same steps as those in the regular devices (with ETL), except for the ETL deposition step, which was omitted. Hence, the reference devices have the following structure: PET/ITO/MAPbI_3_/Spiro-MeOTAD/Au.

### 2.5. Characterization

A transmission electron microscope (TEM, JEOL JEM - 1400 Plus, Tokyo, Japan) was used to estimate the particle size and crystallinity of the obtained SnO_2_ QDs. An X-ray diffractometer (XRD, SmartLab Rigaku, Austin, TX, USA) was used to examine the crystal structure of the SnO_2_ QDs (using Cu K_α_ = 1.5418 Å). A scanning electron microscope (SEM, Zeiss Crossbeam 540, Oberkochen, Germany) was used to obtain the cross-section images of the fabricated devices and to study the morphology of the prepared ETLs. The optical transmittance of the samples was measured using a UV-Vis spectrometer (Lambda 1050, PerkinElmer, Waltham, MA, USA). The performance of the fabricated devices was characterized using a solar simulator (Oriel Sol3A, Newport, Irvine, CA, USA) and a Keithley 2400 source (Keithley, Solon, Ohio, USA). The current density-voltage (*J-V*) curves of the devices were recorded at a scan rate of 400 mV s^−1^ between −0.2 V and +1.2 V. The device area was 0.1 cm^2^, and a mask with an active area of 0.03 cm^2^ was used to measure the device *J-V* curves. An external quantum efficiency (EQE) measurements system (ORIEL IQE-200, Newport, Irvine, CA, USA) was used to record the EQE spectra of the devices.

## 3. Results and Discussion

### 3.1. Crystallinity of SnO_2_ QDs

The XRD measurements were performed on as-prepared SnO_2_ QDs to analyze their structure. Figure 3 shows an XRD pattern of as-prepared SnO_2_ QDs. The XRD pattern shows the diffraction peaks at 2θ = 26.7°, 34.05°, 38.24°, 52.1°, 58.07°, 62.5°, and 65.00°, which correspond to the reflections from the (110), (101), (200), (211), (002), and (301) lattice planes of SnO_2_, respectively. All of the peaks match well with JCPDS Data Card No. 77-0452, indicating that the current structure is for SnO_2_, which exists in the tetragonal rutile crystalline phase. The crystal lattice parameters were calculated to be a = 0.4739 nm and c = 0.3221 nm (JCPDS card # 77-0452) [51,52,53]. The estimated average crystallite size for the SnO_2_ QDs is around 2.4 nm, which was calculated from the two major peaks at (110) and (211) using Scherrer’s equation (see Appendix A) [38,39]. The full width at half-maximum (FWHM) values for the (110) and (211) peaks are about 3.6° and 3.5°, respectively.

### 3.2. Transmission Electron Microscopy

A TEM analysis is conducted to obtain the electron micrographs of as-prepared SnO_2_ QDs and to estimate their particle size. The average particle size estimated from the TEM images (see Figure 4 and Appendix A) is around 2.4 nm, which is in good agreement with the crystallite size determined using the XRD data shown in Figure 3 [48]. The TEM image also indicates that the prepared particles have a uniform size distribution (Appendix A) and a crystalline structure, which is evident from the fringes shown in Figure 4b. The estimated value of d-spacing from the TEM image of the as-prepared SnO_2_ QDs is around 0.33 nm (see the inset image in Figure 4b). This corresponds to the {110} planes of SnO_2_. Since the size of the prepared SnO_2_ QDs is smaller than the Bohr exciton radius (around 2.7 nm), it is reasonable to state that the prepared particles are QDs of SnO_2_ [48,54].

### 3.3. Optical Analysis

The investigation of the optical properties of the ETLs prepared using SnO_2_ inks with different SnO_2_ QD concentrations was carried out using a UV-Vis spectrometer. Figure 5 shows the measured transmittance spectra of the slot-die coated ETLs on glass substrates for the 250–900 nm wavelength range. A comparison of the UV-Vis spectra for different ETLs shows that the transmittance of the ETL made with a 1 wt% SnO_2_ QD ink is the highest and is well above 90% throughout the entire visible spectrum. The second highest transmittance is for the ETL made with a 2 wt% SnO_2_ QD ink, which is slightly higher than the one made with a 1.5 wt% SnO_2_ QD ink. The UV-Vis transmittance of the ETL made with a 3 wt% SnO_2_ QD ink is slightly lower than the one made with the 1.5 wt% SnO_2_ QD ink. The lowest transmittance is observed for the ETL made with a 2.5 wt% SnO_2_ QD ink. Overall, the UV-Vis transmittance measurements for the ETLs made with the 1, 2, and 3 wt% SnO_2_ QD inks show that, with an increase in the concentration of SnO_2_ QDs in the inks, the obtained ETL films become thicker. This is expected, since inks with a higher SnO_2_ QD content should produce thicker films. The observed minor inconsistencies with the ETLs made with the 1.5 and 2.5 wt% SnO_2_ QD inks could be associated with some small deviations in the ink preparation and deposition steps, which may influence the thickness of the resulting films. It is noteworthy to point out that the transmittance of all the ETLs is above 90%, which is beneficial from the PSC device perspective. Such high light transmittances of the obtained ETLs allow more photons to pass into the perovskite layer and enable the generation of more charge carriers in the complete devices [55,56,57,58].

### 3.4. Scanning Electron Microscopy

The SEM imaging is used to determine the thickness of the device functional layers. Figure 6a shows the cross-section of an ETL on glass/FTO made using a 2 wt% SnO_2_ QD ink. The thickness of the SnO_2_ QD ETL is about 25 nm. The cross-section SEM images of ETLs made using the 1, 1.5, 2.5, and 3 wt% SnO_2_ QD inks are shown in Appendix A. Figure 6b shows all three functional layers of a PSC slot-die coated on glass/FTO. Here, the ETL is about 21–25 nm thick (see inset image), while the MAPbI_3_ and the HTL have thicknesses of around 600 and 650 nm, respectively. Figure 6c shows a top view SEM image of a bare PET/ITO substrate, in which the crystallites of the ITO layer are clearly visible. Figure 6d shows the top view SEM image of a 2 wt% SnO_2_ QD ETL slot-die coated on a PET/ITO substrate. The image indicates that the layer of SnO_2_ QDs completely covers the surface of the PET/ITO substrate, as no visible pinholes can be observed from the ETL through which the surface of the ITO can be seen. By comparing Figure 6c,d, the SnO_2_ QD ETL prepared by our technique can uniformly cover the surface of the PET/ITO substrate. A uniform coverage of the TCO layer by an ETL is necessary for good device operation.

### 3.5. Device Analysis

Flexible and printed PSCs on PET/ITO substrates without and with ETLs, prepared using SnO_2_ QD solutions with different concentrations (0, 1, 1.5, 2, 2.5, and 3 wt%), were fabricated to rule out the suitable concentration of SnO_2_ QDs in the ETL inks. For this, devices are fabricated for each concentration of SnO_2_ QDs in the ETL inks. Hereon, the ETLs prepared with SnO_2_ QDs with 0, 1, 1.5, 2, 2.5, and 3 wt% will be referred to as E0, E1, E2, E3, E4, and E5, respectively. It should be noted that E0 refers to bare ITO. The prepared devices are tested, and a statistical analysis of the performance parameters of the top 40 well-performing devices for different ETLs (E0, E1, E2, E3, E4, and E5) is carried out.

Figure 7 shows the statistical data for the solar cell parameters (obtained from the reverse scans) such as open-circuit voltage (V_oc_), short-circuit current density (J_sc_), fill factor (FF), PCE, series resistances (R_s_), and shunt resistances (R_Sh_) presented for devices with E0, E1, E2, E3, E4, and E5 ETLs. Figure 7a shows that the average J_sc_ values increase from E0 to E4. For E0, the J_sc_ value is around 1.12 mA cm^−2^, which, for E1, increases up to 7.7 mA cm^−2^. For E2, the J_sc_ value is 12.95 mA cm^−2^, which is higher than that in E1 by around 40%. The average J_sc_ value for E3 (13.28 mA cm^−2^) is only slightly higher than that in E2. The average J_sc_ value of E4 is about 14.31 mA cm^−2^, which is the highest among the compared devices. As for E5, there is a small drop in the average J_sc_ value as compared to E4. Figure 7b shows that there is also a continuous increase in the average V_oc_ values from E0 to E3. The average V_oc_ value in E3 is around 0.93 V, which is ~2.5 times higher than that in E0 and ~13% higher than that in E1. Afterwards, a continuous decrease in the average V_oc_ values was observed for the devices with higher concentrations of SnO_2_ QDs in the ETL inks (E4 and E5).

The observed variations of the V_oc_ and J_sc_ values with the change in the concentration of SnO_2_ QDs in the ETL inks might be due to the impact of the ETL morphology and thickness on the carrier recombination processes and series resistance in the devices (see Table 1) [59]. With an increase in the concentration of SnO_2_ QDs in the ETL ink, the resulting devices show higher R_S_ values, as depicted in Figure 7e. The increase in R_Sh_ values with an increase in the concentration of SnO_2_ QDs in the ETL inks (see Figure 7f) might indicate that the resulting ETL becomes more compact and has fewer pinholes. This helps to reduce charge carrier back-reaction, which is possibly the reason for the initial increase in the average V_oc_ and J_sc_ values (E0 to E3). The following decrease in the average J_sc_ value in E5 might originate from the suboptimal thickness of the ETL, which decreases the transmission of photons through the ETL (see Figure 5). As for the average V_oc_ values in E4 and E5, it might be due to the higher charge recombination at the ETL/perovskite interface due to high the R_S_ in the devices and the lower transmittance of the ETL films as compared to E3 [60].

Figure 7c depicts that there is a steep increase in the average FF values from E0 to E3, followed by a slow decrease for E4 and E5. This trend is somewhat similar to the ones observed in V_oc_ and J_sc_. The average FF value has a remarkable increase of around 37% when the SnO_2_ QD concentration in the ETL ink is doubled (from 1 wt% (E1) to 2 wt% (E3)). Generally, FF is a complex phenomenon. It depends on many device parameters and processes taking place in solar cells. Here, the initial increase in the average FF values from E0 to E3 could be associated with a decrease in the charge carrier recombination and, possibly, an increase in the electrical conductivity of the ETLs due to an increase in their thickness [61,62,63]. For E4 and E5, the decrease in FF might be due to an increase in R_S_ in the devices [60].

The trends in the V_oc_, J_sc_, and FF values determine the final trend for the overall variation of the average PCE with the change in the concentration of SnO_2_ QDs in the ETL inks. As expected and consistent with the variations in the average values of V_oc_, FF, and, to a certain extent, J_sc_, the average PCE values of the devices increase from E0 to E3 and then decrease. The average PCE in E3 is almost three times higher than that in E1. This is mostly due to the significant increases in the average J_sc_ (by a factor of ~1.8) and FF (by a factor of ~1.5) values when the SnO_2_ QD concentration in the ETL ink is doubled (from 1 wt% (E1) to 2 wt% (E3)). All the devices with ETLs (E1, E2, E3, E4, and E5) have significantly higher performances compared to the devices without any ETL (E0). This indicates the necessity of ETLs to obtain a good performance of PSC devices. A detailed overview of the solar cell parameters is presented in Table 1. We note that most of the metal oxide (e.g., ZnO, SnO_2_, and NiO) inks that are commercially available and used in printed optoelectronic devices also have a metal oxide solid content at around 2–2.5 wt%. In light of this, the trends observed in this work appear reasonable and somewhat consistent with the commercially available inks [64,65,66,67].

The external quantum efficiency (EQE) measurements are also performed on the champion device with E3 ETL to obtain its spectral response. Figure 8b shows that the device has an EQE spectrum that is typical for the MAPbI_3_-based PSCs [68,69]. The device shows more or less stable EQE values at around 50% throughout the visible spectrum. The integrated J_sc_ value obtained from the EQE spectrum of the champion device is around 11 mA cm^−2^ (Figure 8b). This is comparable to the J_sc_ value of the device extracted from the *J-V* curve for the forward scan direction (see Table 2). The EQE spectra of the ’champion devices’ with E0, E1, E2, E4, and E5 ETLs are presented in Appendix A.

In order to study the mechanical stability of flexible PSCs, the fabricated device with E3 ETLs is subjected to a repeated cycle of compressive bending to a radius of around 1 cm, and their solar cell performance is measured after every 10 bending cycles. Overall, the devices are subjected to 50 bending cycles. Figure 9 shows the variations in the solar cell parameters of a device with E3 ETL upon bending. After the first 10 bending cycles, the Jsc value of the device decreases by 45%, and the FF value decreases by 37%. These fast decays in the Jsc and FF values after the first 10 bending cycles could mainly be associated with the deterioration of the structural integrity of the device due to the appearance of cracks in the ITO layer [70,71,72,73,74]. This significantly increases the device series resistance. Between the 10th and 50th bending cycles, the FF value of the device decays rather slowly and monotonously. A somewhat similar but slightly faster decay of the device’s Jsc value is observed for the same bending cycle range. These decays could be due to the further and more complex deterioration (e.g., micro-delamination of the device functional layers, formation of micro-cracks and micro-voids in the device[s functional layers, etc.) of the device’s structural integrity [75,76,77]. As for the device’s Voc, there is almost a linear decay trend with the increase in the number of bending cycles. This indicates the formation of many defects (microcracks) in the perovskite layer, which increases the rate of charge carrier recombination in the layer and leads to a corresponding decrease in the Voc and Jsc values of the devices [78,79,80,81]. These trends in the behavior of the Jsc, FF, and Voc during the bending test result in a fast decay of the device’s PCE after the first 10 bending cycles (a decrease from 7% to 2.5%) and a slower decay between the 10th and 50th bending cycles. After the 50th bending cycle, the PCE of the device becomes around 0.84%. This is smaller (by a factor of ~8) than the initial PCE value (7%) of the device. The mechanical stability could be further improved by the modification of the functional layers and their interfaces in PSCs as well as the application of more durable TCO layers [82].

## 4. Conclusions

In summary, the synthesis and application of SnO_2_ QDs as ETLs for flexible and printed PSCs based on PET/ITO substrates are demonstrated. SnO_2_ QDs with diameters around 2.4 nm are obtained using a solvothermal synthesis method. The QDs of SnO_2_ in the prepared inks are crystalline and well-dispersed. The device functional layers (ETL, perovskite, and HTL) are deposited using a slot-die coating method under ambient air conditions. In order to optimize the printed PSCs, the water-based inks with different SnO_2_ QD concentrations for preparing different thicknesses of ETLs are investigated. The devices with ETLs obtained by using a 2 wt% SnO_2_ QD ink show the best performance (the average PCE for the reverse scan direction is around 7.23%). The champion device, which shows a PCE value of around 10%, is also obtained with the ETL prepared using the 2 wt% SnO_2_ QD ink. The mechanical stability of a device with the 2 wt% SnO_2_ QD ETL is investigated by subjecting the device to a bending test. The initial fast degradation of the performance of the device is attributed to the cracking of the ITO layer in the device, whereas the further slower decay is a result of more complex deterioration in the structure of the device. The findings of the work can be useful in developing new electronic inks for optoelectronic applications and in establishing the advanced low-temperature manufacturing protocols for the printing of optoelectronic devices on heat-sensitive substrates.

## Figures and Tables

**Figure 1 nanomaterials-12-02615-f001:**
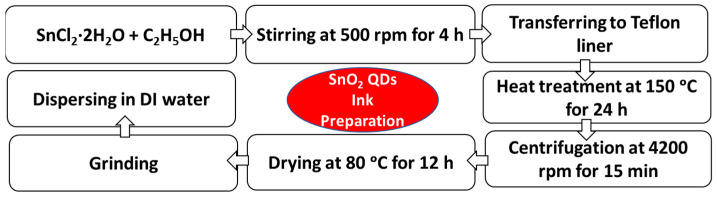
Preparation of ETL ink with SnO_2_ QDs.

**Figure 2 nanomaterials-12-02615-f002:**
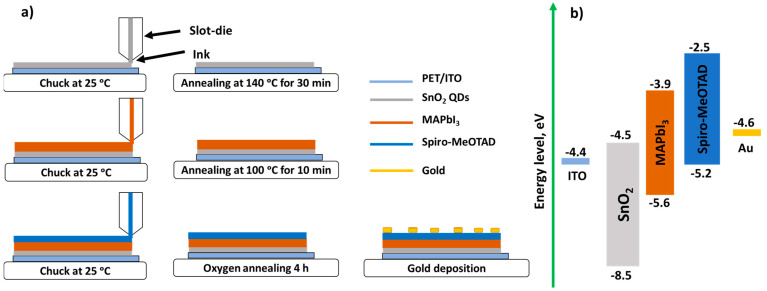
(**a**) Schematic diagram of the device fabrication process. (**b**) Energy level diagram for PSCs.

**Figure 3 nanomaterials-12-02615-f003:**
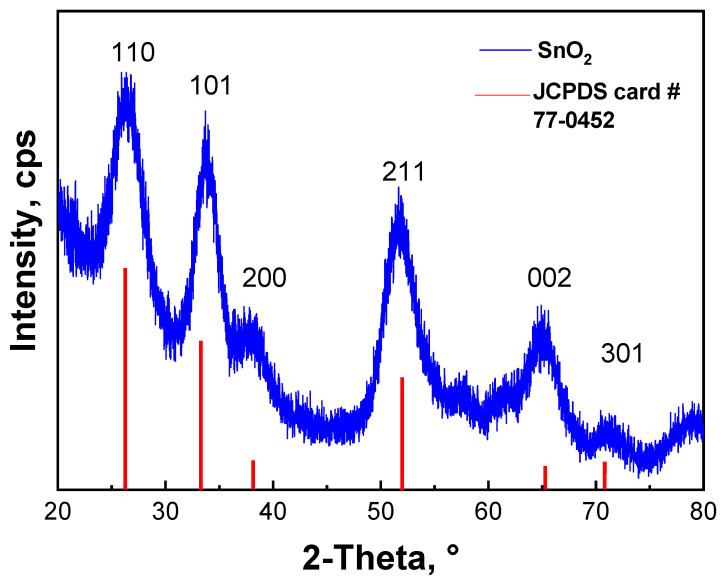
XRD pattern of as-prepared SnO_2_ QDs.

**Figure 4 nanomaterials-12-02615-f004:**
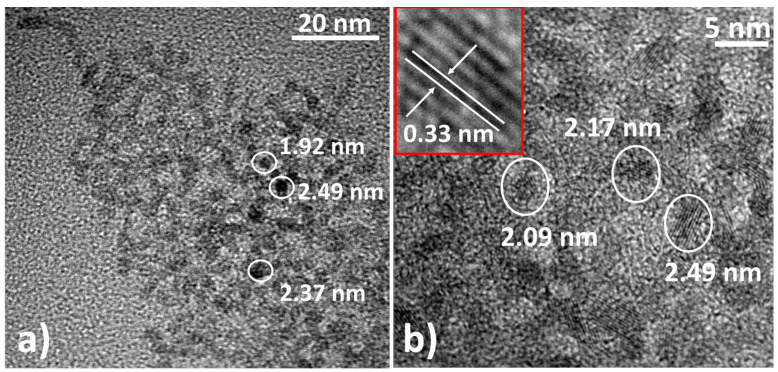
TEM micrographs of SnO_2_ QDs synthesized via a solvothermal route at a magnification of (**a**) 40 kX and (**b**) 1500 kX. The inset image in (**b**) shows the fringes and d-spacing of SnO_2_ QDs.

**Figure 5 nanomaterials-12-02615-f005:**
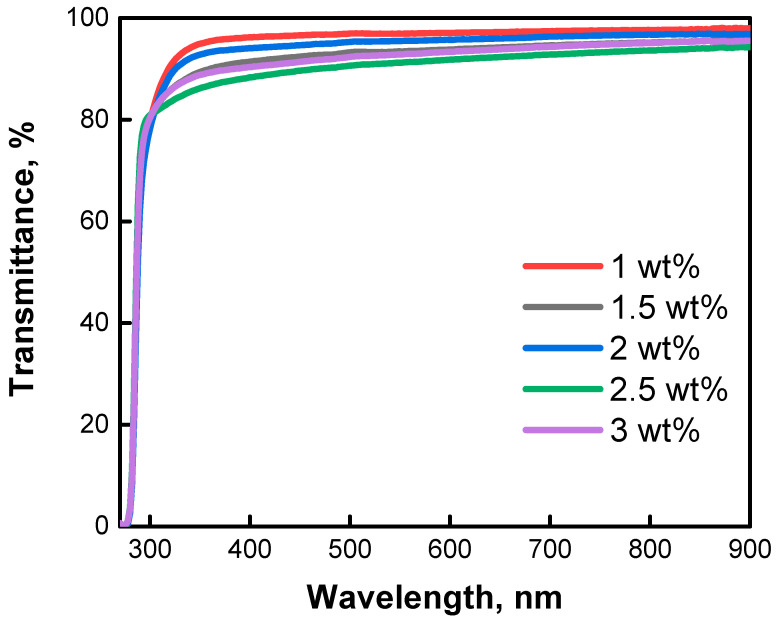
UV-Vis transmittance spectra of ETLs on glass substrates prepared using ETL inks with different SnO_2_ QD concentrations.

**Figure 6 nanomaterials-12-02615-f006:**
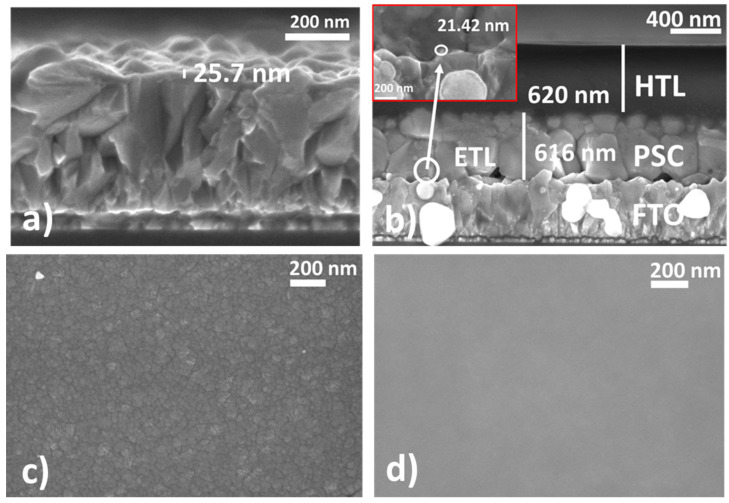
(**a**) SEM cross-section view of the ETL with SnO_2_ QDs on glass/FTO. (**b**) SEM cross-section view of slot-die coated device functional layers on glass/FTO (inset shows the presence of ETL between the FTO and perovskite layers. (**c**) SEM top view image of the PET/ITO substrate (magnification at 100 kX). (**d**) SEM top view image of the SnO_2_ QD ETL on the PET/ITO substrate (magnification at 100 kX).

**Figure 7 nanomaterials-12-02615-f007:**
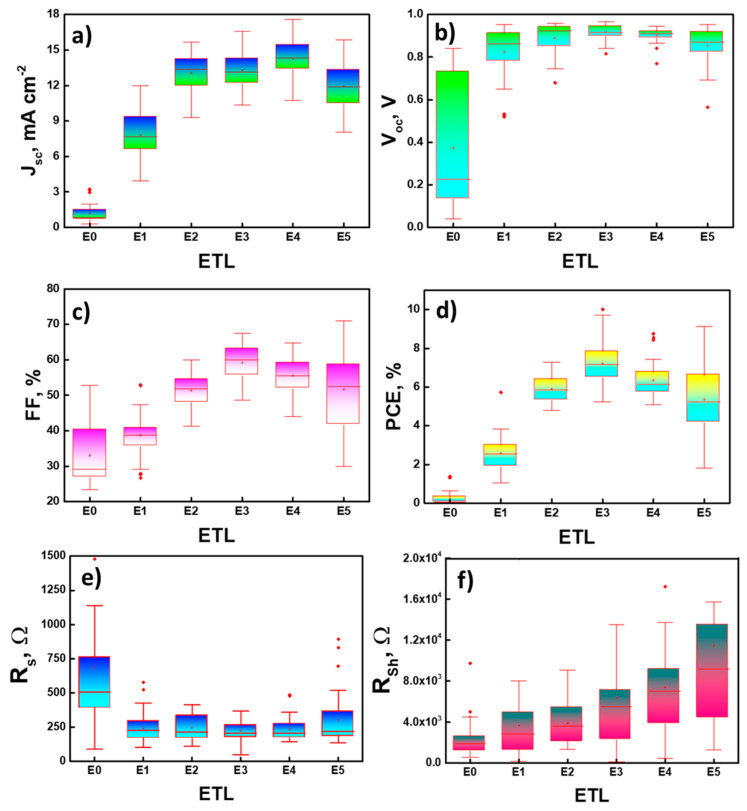
Comparison of the average values of (**a**) J_sc_, (**b**) V_oc_, (**c**) FF, (**d**) PCE, (**e**) R_S_, and (**f**) R_Sh_ of printed and flexible PSCs for different SnO_2_ ETLs prepared using SnO_2_ QD inks with different QD concentrations.

**Figure 8 nanomaterials-12-02615-f008:**
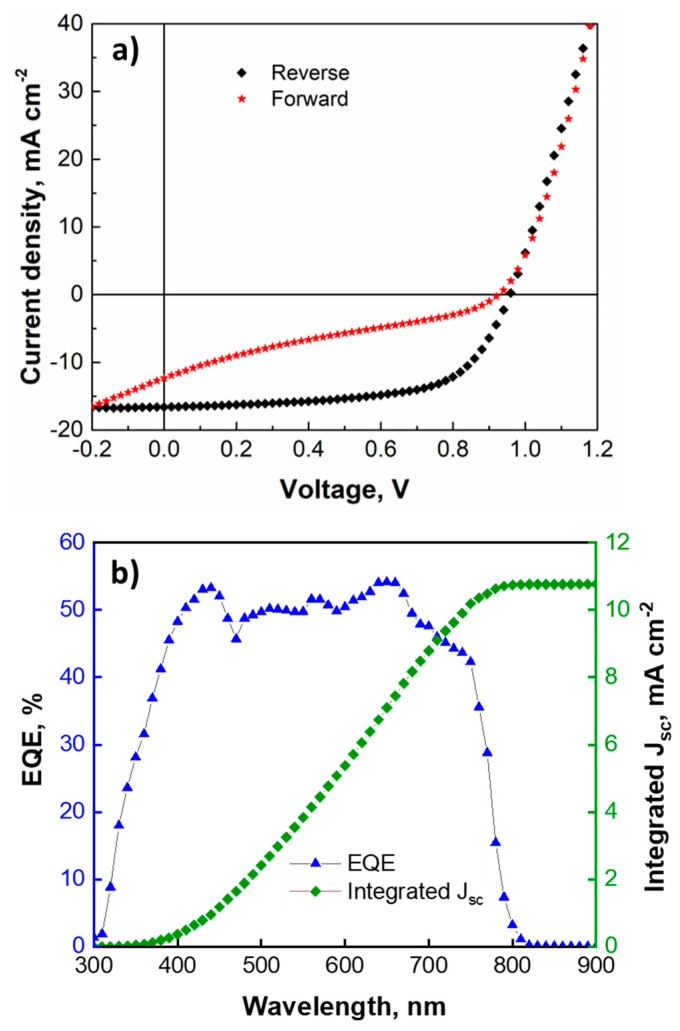
(**a**) *J-V* curves and (**b**) EQE spectrum with an integrated J_sc_ curve for the champion device.

**Figure 9 nanomaterials-12-02615-f009:**
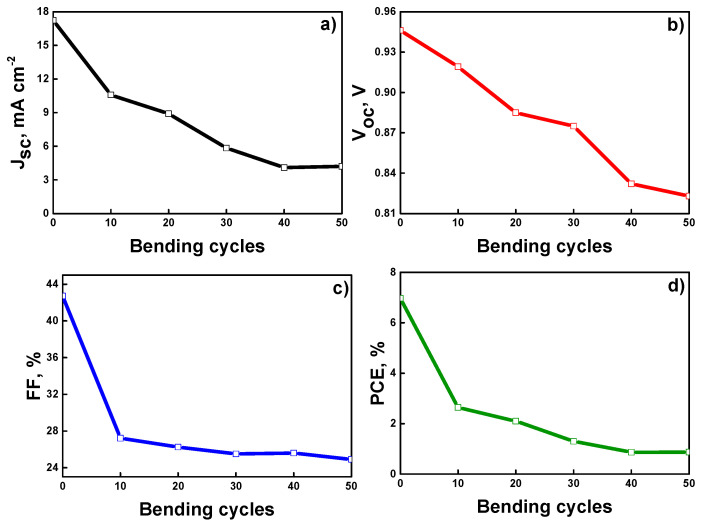
Variation in the (**a**) J_sc_, (**b**) V_oc_, (**c**) FF, and (**d**) PCE of the device upon bending.

**Table 1 nanomaterials-12-02615-t001:** Statistical analysis of the performance parameters of the top 40 well-performing devices for different ETLs.

ETLs.	Scan Direction	Jsc, mA cm^−2^	Voc, V	FF, %	PCE, %	R_S_, Ω	R_Sh_, kΩ
E0 (ref.)	Reverse Forward	1.15 ± 0.113 0.94 ± 0.102	0.37 ± 0.046 0.33 ± 0.048	33.06 ± 1.326 27.95 ± 0.484	0.21 ± 0.050 0.12 ± 0.027	689.6 ± 119 994.8 ± 252.2	4.6 ± 2.4 16.6 ± 10.4
E1 (1 wt%)	Reverse Forward	7.72 ± 0.293 5.56 ± 0.264	0.82 ± 0.017 0.72 ± 0.021	38.69 ± 0.849 28.85 ± 0.723	2.52 ± 0.164 1.19 ± 0.096	238.4 ± 16.6 447.8 ± 121.1	3.7 ± 0.5 3.9 ± 0.4
E2 (1.5 wt%)	Reverse Forward	12.95 ± 0.491 10.08 ± 0.572	0.89 ± 0.013 0.82 ± 0.014	51.15 ± 1.576 36.29 ± 1.453	5.84 ± 0.338 2.99 ± 0.23	246.6 ± 14.1 230.7 ± 11.4	3.9 ± 0.3 3.7 ± 0.3
E3 (2 wt%)	Reverse Forward	13.28 ± 0.22 8.52 ± 0.35	0.93 ± 0.01 0.88 ± 0.01	59.22 ± 0.75 28.27 ± 0.97	7.23 ± 0.18 2.15 ± 0.12	222.7 ± 11.2 221.3 ± 10.3	6.3 ± 1.4 3.3 ± 0.3
E4 (2.5 wt%)	Reverse Forward	14.31 ± 0.63 10.34 ± 0.64	0.92 ± 0.01 0.85 ± 0.01	53.10 ± 1.82 32.50 ± 1.44	6.98 ± 0.46 3.08 ± 0.32	232.6 ± 12.9 223.5 ± 15.4	7.4 ± 1.1 4.2 ± 0.4
E5 (3 wt%)	Reverse Forward	11.76 ± 0.30 7.52 ± 0.30	0.83 ± 0.02 0.75 ± 0.02	50.62 ± 1.90 25.41 ± 0.77	5.16 ± 0.31 1.50 ± 0.12	297.2 ± 28.6 271.0 ± 27.4	11.7 ± 1.7 5.4 ± 0.8

Statistically, E3 ETL is found to be the most suitable to obtain well-performing devices. It is also noteworthy that the highest performing device (champion device) is also obtained using E3 ETLs. Figure 8a shows the *J-V* curves of the champion device with E3 ETL. The V_oc_, J_sc_, and FF of the champion device for the reverse scan are around 0.96 V, 17 mA cm^−2^, and 63%, respectively. This results in a device PCE of around 10%. The solar cell parameters obtained from both the forward and reverse *J-V* scan directions are presented in Table 2. The *J-V* curves of the ‘champion devices’ with E0, E1, E2, E4, and E5 ETLs and the corresponding solar cell parameters are presented in Appendix A.

**Table 2 nanomaterials-12-02615-t002:** Solar cell performance parameters for the champion device.

Scan Direction	J_sc_, mA cm^−2^	V_oc_, V	FF, %	PCE, %	Integrated J_sc_, mA cm^−2^
Reverse	16.6	0.958	62.9	10.006	10.76
Forward	12.3	0.925	25.4	2.906	

## Data Availability

The data presented in this study are available on request from the corresponding author.

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
