# Peer review of "Solution-Processed SnO2 Quantum Dots for the Electron Transport Layer of Flexible and Printed Perovskite Solar Cells"

_nanomaterials, 2022, doi:10.3390/nano12152615_

Round 1

Reviewer 1 Report

This manuscript reports “Solution-Processed SnO2 Quantum Dots for Electron Transport Layer of Flexible and Printed Perovskite Solar Cells”. Metal oxides such as TiO2, ZnO, and SnO2 are the most prominent candidates as an ETL of PSCs, because of their high electron mobility. In particular solution-processed metal oxide films can be introduced as an ETL involving “optical spacers” that increase the absorption of light. Although SnO2 has been studied for decades in solar cell communities, it is still necessary to improve long-term stability and processibility of SnO2 interlayers. Therefore, this work might be of interest for publication if the authors can clarify the following questions:

1.     I think that the energy level diagram of prepared devices would be helpful for the readers. Please add the energy level diagram including ETLs.

2.     I suggest the authors to provide a device performance of PSC without ETL (reference device) in figure 7 and table 2.

3.     In table 2, the series and sheet resistance of devices should be calculated, and their effect on the device performance should be discussed.

4.     The light absorption efficiencies of photoactive layers can significantly influence on the PSC performances. Therefore, the light absorption properties and EQE spectra for the reference (without ETL) and E1-E5 devices should be displayed and discussed.

5.     The authors claimed that the “flexible perovskite solar cells” based on solution-processed SnO2 QD ETLs. Accordingly, it is suggested to provide bending test data exhibiting flexibilities of prepared PSCs. The detailed method and results can be referred to the following reference: Critical review of recent progress of flexible perovskite solar cells. Mater. Today 2020, 39, 66.

6. Please add some references relevant to the screen printing, blade coating, inkjet printing in page 1, line 38-40

Author Response

Detailed Response Letter

Reviewer #1

Reviewer’s general comment:

This manuscript reports “Solution-Processed SnO2 Quantum Dots for Electron Transport Layer of Flexible and Printed Perovskite Solar Cells”. Metal oxides such as TiO2, ZnO, and SnO2 are the most prominent candidates as an ETL of PSCs, because of their high electron mobility. In particular solution-processed metal oxide films can be introduced as an ETL involving “optical spacers” that increase the absorption of light. Although SnO2 has been studied for decades in solar cell communities, it is still necessary to improve long-term stability and processibility of SnO2 interlayers. Therefore, this work might be of interest for publication if the authors can clarify the following questions:

Authors’ response to the Reviewer’s general comment:

We thank the Reviewer for his/her time and effort in reviewing our manuscript and we are grateful to the Reviewer for appreciation of our work. We also would like to thank the Reviewer for providing us with useful comments and the opportunity to revise our manuscript.

Reviewer’s comment #1:

I think that the energy level diagram of prepared devices would be helpful for the readers. Please add the energy level diagram including ETLs.

Authors’ response to comment #1:

As per suggestion of the Reviewer, an energy level diagram of prepared devices (see Figure 2b) is now added in the revised manuscript. The associated changes in the main text of the manuscript are highlighted in yellow (see pp. 5).

Reviewer’s comment #2:

I suggest the authors to provide a device performance of PSC without ETL (reference device) in figure 7 and table 2.

Authors’ response to comment #2:

As per suggestion of the Reviewer, reference devices without ETL is prepared and the device performances are tested. The data on the performance of reference PSCs without ETL is now presented in Figure 7 and Table 1. The associated changes in the main text of the manuscript are highlighted in yellow (see pp. 5, 9-12).

Reviewer’s comment #3:

In table 2, the series and sheet resistance of devices should be calculated, and their effect on the device performance should be discussed.

Authors’ response to comment #3:

As per suggestion of the Reviewer, the series and sheet resistance of prepared devices are calculated and presented in Table 1 and their effect on the device performance is discussed in the revised manuscript. The associated changes in the main text of the manuscript are highlighted in yellow (see pp. 9-12).

Reviewer’s comment #4:

The light absorption efficiencies of photoactive layers can significantly influence on the PSC performances. Therefore, the light absorption properties and EQE spectra for the reference (without ETL) and E1-E5 devices should be displayed and discussed.

Authors’ response to comment #4:

As per suggestion of the Reviewer, EQE spectra (see Figure S4 in Supporting Information) as well as the J-V curves with the indication of solar cells parameters (see Figure S3 in Supporting Information) of the devices without (E0) and with (E1, E2, E4, and E5) ETLs are now presented in Supporting Information of the revised manuscript. The associated changes in the main text of the revised manuscript are highlighted in yellow (see pp. 12).

Reviewer’s comment #5:

The authors claimed that the “flexible perovskite solar cells” based on solution-processed SnO2 QD ETLs. Accordingly, it is suggested to provide bending test data exhibiting flexibilities of prepared PSCs. The detailed method and results can be referred to the following reference: Critical review of recent progress of flexible perovskite solar cells. Mater. Today 2020, 39, 66.

Authors’ response to comment #5:

We thank the Reviewer for a useful suggestion. In the revised manuscript, we now include the bending test results performed on prepared devices (see Figure 9). A new paragraph dedicated to describing and discussing the bending test results has been introduced in the “Results and Discussion” section of the revised manuscript. The reference suggested by the Reviewer is also cited in the revised manuscript (see Ref. 45). The associated changes in the main text of the revised manuscript are highlighted in yellow (see pp. 14 and 18).

Reviewer’s comment #6:

Please add some references relevant to the screen printing, blade coating, inkjet printing in page

1, line 38-40.

Authors’ response to comment #6:

As per suggestion of the Reviewer, the additional references for screen printing, blade coating, and inkjet printing are now cited in the revised manuscript [Ref. 11-14]. The associated changes in the main text are highlighted in yellow (see pp. 1 and 16).

Reviewer 2 Report

1. Regarding the ETL research, I suggest author can review and cite RSC Advances 2018, 8, 12526-12534 in introduction. It report using organic material as the ETL. That would help readers to know more about ETL.

2. Please write clearly the device structure in experiment. Demenstrate the materials in each layer.

3. What is the size of each device cell ?

4. How the SnO2 QD concentration affect the size ? (TEM images) 

5. Author report using the flexible substrate PET/ITO. Does any device testing for bending the substrate  ?

Author Response

Detailed Response Letter

Reviewer #2

Authors’ general response to the Reviewer:

We thank the Reviewer for his/her time and effort in reviewing our manuscript and we are grateful to the Reviewer for providing us with useful comments and the opportunity to revise our manuscript.

Reviewer’s comment #1:

Regarding the ETL research, I suggest author can review and cite RSC Advances 2018, 8, 12526-12534 in introduction. It report using organic material as the ETL. That would help readers to know more about ETL.

Authors’ response to comment #1:

As per suggestion of the Reviewer, the reference mentioned is now reviewed and cited in the revised manuscript [Ref. 30]. The associated changes in the main text of the manuscript are highlighted in yellow (see pp. 2 and 17).

Reviewer’s comment #2:

Please write clearly the device structure in experiment. Demonstrate the materials in each layer.

Authors’ response to comment #2:

As per suggestion of the Reviewer, the device structure and materials used in fabrication of devices are clearly indicated in the revised manuscript. The associated changes in the main text of the manuscript are highlighted in yellow (see pp.4-5).

Reviewer’s comment #3:

What is the size of each device cell?

Authors’ response to comment #3:

The device area is 0.1 cm2 and a mask with an active area of 0.03 cm2 is used to measure the device J-V curves. This information is now reflected in the revised manuscript. The associated changes in the main text of the manuscript are highlighted in yellow (see pp. 5).

Reviewer’s comment #4:

How the SnO2 QD concentration affect the size? (TEM images) 

Authors’ response to comment #4:

We believe that the Reviewer wanted to ask how the concertation of the Sn precursor affects the size of SnO2 QDs. The protocol used in our work for the synthesis of SnO2 QDs is similar to the one reported by Zhu et al. (doi:10.1088/0957-4484/17/9/052; [Ref. 48]). We have not varied the concentration of the Sn precursor in our synthesis protocol since the one we used already produced the desired uniform size (around 2.4 nm) of SnO2 nanoparticles, which is in the QD range. Therefore, the variation of the Sn precursor solution on the size of SnO2 QDs is not critical in this work.

Reviewer’s comment #5:

Author report using the flexible substrate PET/ITO. Does any device testing for bending the substrate?

Authors’ response to comment #5:

We thank the Reviewer for a useful comment. In the revised manuscript, we now include the bending test results performed on prepared devices (see Figure 9). A new paragraph dedicated to describing and discussing the bending test results has been introduced in the main text of the manuscript. The associated changes in the main text of the revised manuscript are highlighted in yellow (see pp. 14).

Reviewer 3 Report

The authors synthesized SnO2 and studied the concentration of SnO2 aqueous solution on the performance of perovskite solar cells. Sn-based ETL is an important topic in the development of perovskite solar cells and sandwich structure optoelectronic devices. There have been many reports on SnO2, but the work here is not a very good one. Hope the comments below would be helpful.

1 The authors claimed their SnO2 can work well with low processing temperature, however, they set the annealing temperature of SnO2 to 140 oC carefully and didn’t discuss the performance of SnO2 with higher annealing temperature. It’s better to compare the SnO2 with different annealing temperatures, so the audience may know the low-temperature processing nature of the SnO2 reported here.

2 It is well known that the concentration of SnO2 aqueous solution can affect the compactness and thickness in the slot-die process, while there are few direct data on that.

3 It suggests showing the key parameters in the calculation of Scherrer’s equation, such as the FWHM of the peak. Meanwhile, please correct the typos in L201.

4 The size of the nanoparticles was determined by TEM, but the data in figure 4 cannot show the average size of the particle is 2.4nm. Particle size distribution is suggested.

5 The description of UV-vis seems slightly to deviate from the truth. The 2 wt% sample shows a little higher transmittance than other samples after 550nm. And it is interesting that the solution concentration can lead to a sudden change in transmittance (the spectrum of 2 wt% sample is obviously different than others), so it’s better to give a reasonable explanation.

6 it’s hard to observe the thickness of the ETL from the SEM image in Figure 6. Meanwhile, it’s hard to believe that such a thin film can be coated evenly onto a such rough surface.

Author Response

Detailed Response Letter

Reviewer #3

Reviewer’s general comment:

The authors synthesized SnO2 and studied the concentration of SnO2 aqueous solution on the performance of perovskite solar cells. Sn-based ETL is an important topic in the development of perovskite solar cells and sandwich structure optoelectronic devices. There have been many reports on SnO2, but the work here is not a very good one. Hope the comments below would be helpful.

Authors’ response to the Reviewer’s general comment:

We thank the Reviewer for his/her time and effort in reviewing our manuscript and we are grateful to the Reviewer for providing us with useful comments and the opportunity to revise our manuscript.

Reviewer’s comment #1:

The authors claimed their SnO2 can work well with low processing temperature, however, they set the annealing temperature of SnO2 to 140 oC carefully and didn’t discuss the performance of SnO2 with higher annealing temperature. It’s better to compare the SnO2 with different annealing temperatures, so the audience may know the low-temperature processing nature of the SnO2 reported here.

Authors’ response to comment #1:

We appreciate the Reviewers insightful comment. The objective of this work is to use already crystalline SnO2 QDs to print crystalline ETLs on heat-sensitive substrates (PET/ITO). Freshly printed SnO2 QD ETLs on PET/ITO substrates obtained in our work still contain some remaining solvent (water). Hence, prior to deposition of the perovskite layer, the remaining water in the ETL need to be removed by annealing the substrates at elevated temperatures. The melting temperature of the PET substrates used in our work is around 150 C. Therefore, it is not recommended to heat these substrates close to or above this temperature. That is why we selected the 140 C annealing temperature as this allows effective evaporation of the remaining solvent from the prepared ETLs and still keep the structural integrity of the PET/ITO substrates. In general, the protocol developed in our work allows the ETL annealing temperatures that are lower than 140 C as long as it is enough to evaporate the remaining water from the ETL films. The ETL annealing temperatures higher than 140 C requires the use of substrates that are more temperature-resilient. This is now reflected in the ‘Introduction’ and ‘Device Fabrication’ section of the manuscript. The associated changes in the main text of the revised manuscript are highlighted in yellow (see pp. 3-4).

Reviewer’s comment #2:

It is well known that the concentration of SnO2 aqueous solution can affect the compactness and thickness in the slot-die process, while there are few direct data on that.

Authors’ response to comment #2:

In the revised manuscript, we now include the SEM cross-section images (see Figure S2 in the Supporting Information) showing the thickness of SnO2 QD ETLs printed on Glass/FTO substrates. Additionally, the thickness and compactness of the obtained ETLs are also discussed based on the UV-Vis transmittance measurements on glass/ETL samples and the device series and shunt resistance data. The associated changes in the main text of the manuscript are highlighted in yellow (see pp. 7-12).

Reviewer’s comment #3:

It suggests showing the key parameters in the calculation of Scherrer’s equation, such as the FWHM of the peak. Meanwhile, please correct the typos in L201.

Authors’ response to comment #3:

In Supporting Information of the revised manuscript, the Scherrer’s equation (Equation 1 in Supporting Information) and its key parameters are presented. The FWHM values for the selected XRD peaks that are used to calculate the particle size using the Scherrer’s equation are now indicated in the main text of the revised manuscript. In addition, the typo mentioned by the Reviewer is now corrected. The associated changes in the main text of the manuscript are highlighted in yellow (see pp. 5).

Reviewer’s comment #4:

The size of the nanoparticles was determined by TEM, but the data in figure 4 cannot show the average size of the particle is 2.4 nm. Particle size distribution is suggested.

Authors’ response to comment #4:

As per suggestion of the Reviewer, the particle size distribution diagram (see Figure S1 in Supporting Information) is now presented in Supporting Information of the revised manuscript. The associated changes in the main text of the revised manuscript are highlighted in yellow (see pp. 6).

Reviewer’s comment #5:

The description of UV-vis seems slightly to deviate from the truth. The 2 wt% sample shows a little higher transmittance than other samples after 550nm. And it is interesting that the solution concentration can lead to a sudden change in transmittance (the spectrum of 2 wt% sample is obviously different than others), so it’s better to give a reasonable explanation.

Authors’ response to comment #5:

We thank the Reviewer for this excellent comment. In the first version of the manuscript, there is a sentence (“Figure 5 shows the measured transmittance spectra of the slot-die coated ETLs on glass substrates for the 300–900 nm wavelength range.”) with a typo. It supposed to read as “Figure 5 shows the measured transmittance spectra of the slot-die coated ETLs on PET/ITO substrates for the 300–900 nm wavelength range”). The UV-Vis measurements are measured taking PET substrate as a reference. Hence, the UV-Vis spectra presented in the first version of the manuscript are for ITO/ETL samples. We apologize for this typo.

In order to correct the mistake and recheck the UV-Vis transmittance of the prepared ETLs, we have performed additional experiments with newly prepared glass/ETL samples. The UV-Vis measurements on the samples are performed by taking the glass substrates as reference. Hence, the updated UV-Vis graph (Figure 5) in the revised manuscript shows the transmittance spectra of the slot-die coated ETLs alone. In the updated UV-Vis transmittance graph, the transmittance of the ETL made with 1 wt% SnO2 QD ink is the highest and well above 90% throughout the entire measurement range. The second highest transmittance is for the ETL made with 2 wt% SnO2 QD ink, which is slightly higher than the one made with 1.5 wt% SnO2 QD ink. The UV-Vis transmit-tance of the ETL made with 3 wt% SnO2 QD ink is slightly lower than the one made with 1.5 wt% SnO2 QD ink. The lowest transmittance is observed for the ETL made with 2.5 wt% SnO2 QD ink. Overall, the UV-Vis transmittance measurements for the ETLs made with 1, 2, and 3 wt% SnO2 QD inks show that with an increase in the concentration of SnO2 QDs in the inks, the obtained ETL films become thicker. This is expected since the inks with more SnO2 QD content produce thicker films. The observed minor inconsistencies with the ETLs made with 1.5 and 2.5 wt% SnO2 QD inks could be associated with some small deviations in the ink preparation and deposition steps, which may influence the thickness of the resulting films. It is noteworthy to point out that the transmittance of all the ETLs is above 90%, which is beneficial from the PSC device perspective. Such high light transmittances of the obtained ETLs afford more photons to pass into the perovskite layer and enabled generation of more charge carriers in the complete devices. This is now reflected in the main text of the manuscript. The associated changes in the main text of the manuscript are highlighted in yellow (see pp. 7-8). 

Reviewer’s comment #6:

It’s hard to observe the thickness of the ETL from the SEM image in Figure 6. Meanwhile, it’s hard to believe that such a thin film can be coated evenly onto a such rough surface.

Authors’ response to comment #6:

In the revised manuscript, we now present the top view SEM images (see Figure 6c and 6d) of a bare PET/ITO and a PET/ITO/ETL samples with higher magnification to demonstrate the even coating of the surface of PET/ITO substrate by the SnO2 QD inks. The text below has now been introduced in the revised manuscript to support this:

            “Figure 6c shows a top view SEM image of a bare PET/ITO substrate, in which the crystal-lites of the ITO layer are clearly visible. Figure 6d shows the top view SEM image of a 2 wt% SnO2 QD ETL slot-die coated on a PET/ITO substrate. The image indicates that the layer of SnO2 QDs completely covers the surface of the PET/ITO substrate as no visible pinholes can be observed from the ETL through which the surface of the ITO can be seen. By comparing the Figure 6c and Figure 6d, the SnO2 QDs prepared by our technique can uniformly cover the surface of the PET/ITO substrate. A uniform coverage of the TCO layer by an ETL is necessary for good device operation.”

In addition, the cross-section SEM images of all ETLs slot-die coated on Glass/FTO substrates are now presented in the revised manuscript (see Figure S2). We hope that new data presented in the revised manuscript suffices the Reviewer’s questions. The associated changes in the main text of the revised manuscript are highlighted in yellow (see pp. 8-9).

Reviewer 4 Report

The authors developed the PSCs on flexible substrates. However, flexibility analysis is missing. Please comment.

I think the low-temperature processing of SnO2 ETL was one of the key points of this study. However, in the introduction, this point is not fully justified. I recommend highlighting the superiority and need of this work, instead of writing a general text about slot-die and other things in the introduction. The comparison should be in such a way that why this study was required when there are many studies already available because this ETL is the most common one in the PSC area. I believe the introduction can be reduced.  

Good optical transmittance is also suitable for indoor PV devices. Please include the suggested important studies regarding this important aspect (https://doi.org/10.1016/j.apsusc.2021.150852; https://doi.org/10.1016/j.jpowsour.2021.230782;  

https://doi.org/10.1002/adfm.202201921).

The authors synthesized the QDs however about their crystallinity, they referred to the reference [40]. Why is it so?

Why there is a significant hysteresis in Figure 8a?

In Table 1, what is meant by the Coating window and related parameters (221, 222-223, etc) in the opposite columns? Also, in Figure 2, what is oxygen annealing? In Figure 5, please check the typo on X-axis. 

Author Response

Detailed Response Letter

Reviewer #4

Authors’ general response to the Reviewer:

We thank the Reviewer for his/her time and effort in reviewing our manuscript and we are grateful to the Reviewer for providing us with useful comments and the opportunity to revise our manuscript.

Reviewer’s comment #1:

The authors developed the PSCs on flexible substrates. However, flexibility analysis is missing. Please comment.

Authors’ response to comment #1:

We thank the Reviewer for a useful comment. In the revised manuscript, we now include the bending test results performed on prepared devices (see Figure 9). A new paragraph dedicated to describing and discussing the bending test results has been introduced in the main text of the revised manuscript. The associated changes in the main text are highlighted in yellow (see pp. 14).

Reviewer’s comment #2:

I think the low-temperature processing of SnO2 ETL was one of the key points of this study. However, in the introduction, this point is not fully justified. I recommend highlighting the superiority and need of this work, instead of writing a general text about slot-die and other things in the introduction. The comparison should be in such a way that why this study was required when there are many studies already available because this ETL is the most common one in the PSC area. I believe the introduction can be reduced.

Authors’ response to comment #2:

We thank the Reviewer for such a useful comment. In the “Introduction” section of the revised manuscript, we now introduced an additional text (see below) highlighting the importance of the low-temperature processing of SnO2 ETLs:

            “The main issue that limits the use of SnO2 ETL for flexible devices is its annealing temperature, which has been found > 150 ᵒC, while the typical flexible substrates such as PET and PEN usually have the temperature limitations in terms of their structural integrity [40,44–46]. One of the possible solutions for this could be the use of already crystalline SnO2 QDs for solution-processed deposition of SnO2 ETLs. Here, the use of already crystalline SnO2 QDs would result not only in crystalline SnO2 ETLs but would also afford to avoid high-temperature processing step, which is crucial for the devices based on plastic substrates. The low-temperature and solution-processed deposition of SnO2 ETL is advantageous from the perspective of flexible and printed optoelectronics.”

We hope this suffices the Reviewer’s comment. The associated changes in the main text are highlighted in yellow (see pp. 3).

Reviewer’s comment #3:

Good optical transmittance is also suitable for indoor PV devices. Please include the suggested important studies regarding this important aspect (https://doi.org/10.1016/j.apsusc.2021.150852; https://doi.org/10.1016/j.jpowsour.2021.230782; https://doi.org/10.1002/adfm.202201921).

Authors’ response to comment #3:

As per suggestion of the Reviewer, the above-mentioned references are now cited [see Ref. 46, 57, and 58]. The associated changes in the main text of the revised manuscript are highlighted in yellow (see pp. 3, 7, 18, and 19).

Reviewer’s comment #4:

The authors synthesized the QDs however about their crystallinity, they referred to the reference [40]. Why is it so?

Authors’ response to comment #4:

The purpose of this reference is to indicate that the obtained value in our experiment is consistent with the findings of other groups.

Reviewer’s comment #5:

Why there is a significant hysteresis in Figure 8a?

Authors’ response to comment #5:

It is well known that passivation of SnO2 ETL can significantly reduce hysteresis in the devices. However, the fabrication of devices in our work is carried out in ambient environment outside the glovebox and the used SnO2 QD ETLs are not passivated prior to deposition of the perovskite layer. We think that this might have been the main reason for getting high hysteresis in our devices. Studying the effect of the passivation of SnO2 QD ETLs on the hysteresis of device will be addressed in our future reports.

Reviewer’s comment #6:

In Table 1, what is meant by the Coating window and related parameters (221, 222-223, etc) in the opposite columns? Also, in Figure 2, what is oxygen annealing? In Figure 5, please check the typo on X-axis.

Authors’ response to comment #6:

The coating window and related parameters (221, 222-223, etc) in Table 1 refer to the parameters used in our FoM VectorSC slot-die coater. We now realize that the parameters indicated in Table 1 is too specific and can be confusing for the general reader or the researchers that employ different slot-die coaters. Therefore, in the revised manuscript we now removed Table 1 to make it easy for the readers to follow the text of the manuscript. As for the “oxygen annealing”, it is a process carried out to dope the Spiro-MeOTAD HTL and enhance its hole transporting properties. In this process, device are in the atmosphere of dry oxygen for few hours. In the revised manuscript, we now clarify this moment (see pp.4). The typo in Figure 5 is now corrected as per suggestion of the Reviewer. The associated changes in the main text are highlighted in yellow (see pp. 4 and 8).

Round 2

Reviewer 1 Report

The authors have addressed all of my comments. I agree to publish this modified version.

Reviewer 4 Report

The manuscript has been significantly improved. However, please check the references. The reference style should follow journal guidelines.